# Protection of H_2_S against Hypoxia/Reoxygenation Injury in Rat Hippocampal Neurons through Inhibiting Phosphorylation of ROCK_2_ at Thr436 and Ser575

**DOI:** 10.3390/ph16020218

**Published:** 2023-01-31

**Authors:** Fang Fang, Ju Sheng, Yan Guo, Jiyue Wen, Zhiwu Chen

**Affiliations:** Department of Pharmacology, School of Basic Medical Sciences, Anhui Medical University, Hefei 230032, China

**Keywords:** hydrogen sulfide, hippocampal neurons, hypoxia/reoxygenation injury, ROCK_2_, phosphorylation sites

## Abstract

Background: H_2_S (hydrogen sulfide) protects cerebral vasodilatation and endothelial cells against oxygen-glucose deprivation/reoxygenation injury via the inhibition of the RhoA-ROCK pathway and ROCK_2_ expression. However, the inhibitory mechanism of H_2_S on ROCK_2_ expression is still unclear. The study aimed to investigate the target and mechanism of H_2_S in inhibition of ROCK_2_. Methods: His-ROCK_2_^wild^ protein was constructed, expressed, and was used for phosphorylation assay in vitro. Liquid chromatography–tandem mass spectrometry (LC–MS/MS) was used to determine the potential phosphorylation sites of ROCK_2_. Recombinant ROCK_2_^wild^-pEGFP-N1, ROCK_2_^T436A^-pEGFP-N1, and ROCK_2_^S575F^-pEGFP-N1 plasmids were constructed and transfected into rat hippocampal neurons (RHNs). ROCK_2_ expression, cell viability, the release of lactate dehydrogenase (LDH), nerve-specific enolase (NSE), and Ca^2+^ were detected to evaluate the neuroprotective mechanism of H_2_S. Results: Phosphorylation at Thr436 and Ser575 of ROCK_2_ was observed by mass spectrometry when Polo-like kinase 1 (PLK1) and protein kinase A (PKA) were added in vitro, and NaHS significantly inhibited phosphorylation at Thr436 and Ser575. Additionally, NaHS significantly inhibited the expression of ROCK_2_ and recombinant proteins GFP-ROCK_2_, GFP-ROCK_2_^T436A^, and GFP-ROCK_2_^S575F^ in transfected RHNs. Compared with empty plasmid, GFP-ROCK_2_^T436A^, and GFP-ROCK_2_^S575F^ groups, NaHS significantly inhibited the release of LDH, NSE, and Ca^2+^ and promoted ROCK_2_ activity in the GFP-ROCK_2_^wild^ group. Thr436 and Ser575 may be dominant sites that mediate NaHS inhibition of ROCK_2_ protein activity in RHNs. Compared with the empty plasmid, GFP-ROCK_2_^T436A,^ and the GFP-ROCK_2_^S575F^ group, NaHS had more significant inhibitory effects on hypoxia/reoxygenation (H/R) injury-induced cell viability reduction and increased LDH and NSE release in the GFP-ROCK_2_^wild^ group. Conclusion: Exogenous H_2_S protected the RHNs against H/R injury via Thr436 and Ser575 of ROCK_2_. These findings suggested that Thr436 and Ser575 may be the dominant sites that mediated the effect of NaHS on protecting RHNs against H/R injury.

## 1. Introduction

Ischemic cerebral stroke is due to an interruption in cerebral blood flow, which is induced by thrombosis or embolism. There was an increased incidence of ischemic cerebral stroke with increasing age worldwide, and the treatment cost of ischemic cerebral stroke is the largest economic burden in East Asia [1]. Restoring blood flow is essential for the treatment of ischemic cerebral stroke, but reperfusion itself may lead to additional neurological function damage and the formation of cerebral infarction, called cerebral ischemia/reperfusion (I/R) injury [2]. Protecting the brain from a cerebral I/R injury is crucial for the treatment of cerebral ischemic stroke. Antithrombotic therapies, including antiplatelet or anticoagulant agents, are recommended for nearly all ischemic stroke patients with no contraindication [3,4]. However, pharmacological approaches against ischemic cerebral stroke remain limited, suggesting the need for new treatments.

Hydrogen sulfide (H_2_S), a novel gaseous signaling molecule, is known as a gasotransmitter in mammals in addition to nitric oxide and carbon monoxide [5]. H_2_S is endogenously synthesized by enzyme cystathionine-γ-lyase, cystathionine-β-synthase, or 3-mercaptopiruvate sulfurtransferases and is involved in the physiological function and pathological process in the brain [6,7]. H_2_S exerts an important neuroprotective effect on the cerebral I/R injury via antioxidant, anti-inflammatory, and anti-apoptotic actions [8], but its precise mechanism is still unclear. 

RhoA, a small GTPase of the Rho family, is involved in the regulation of multiple cellular signal transduction pathways [9]. RhoA and its downstream effector Rho kinase (ROCK) is highly expressed in the nervous system and associated with various neuronal functions and numerous central nervous system diseases [10]. The RhoA-ROCK pathway participates in astrocyte-mediated angiogenesis and neurogenesis; inhibition of the RhoA-ROCK pathway can alleviate neuroinflammation, apoptosis, and oxidative stress, which are beneficial to neural recovery after an ischemic stroke [11]. We previously found that exogenous and endogenous H_2_S had vascular and neuroprotective effects on cerebral I/R-induced dysfunction in mice and rats via the inhibition of the RhoA/ROCK pathway [12,13,14,15,16,17]. It was reported that exogenous and endothelial H_2_S protects rat hippocampal neurons against hypoxia/reoxygenation (H/R) injury by promoting the phosphorylation of RhoA at the Ser188 site [18]. However, there is little research on the effect of H_2_S on ROCK and its underlying mechanism. 

ROCK has two types of isoforms, ROCK_1_ and ROCK_2_, which are identified in cells. ROCK_1_ is mainly expressed in non-neuronal tissues, while ROCK_2_ is more abundantly distributed in the brain and skeletal muscles [19]. A previous study demonstrated that the upregulation of ROCK_2_ aggravated H/R injury in the nerve cells but the downregulation of ROCK_2_ improved the H/R injury [20]. Our recent studies demonstrated that H_2_S promotes the phosphorylation of ROCK_2_ at Tyr722 and inhibits ROCK_2_ activity to protect rat hippocampal neurons from H/R injury [21]. However, as a macromolecular protein with more than 1300 amino acid residues, ROCK_2_ might have multiple dominant sites that can be phosphorylated and affect its activation. Therefore, the present study was designed to investigate the regulation of H_2_S on the phosphorylation of ROCK_2_ at other potential sites except Tyr722 and to explore whether this regulation mediates the activation of ROCK_2_ and the neuroprotective effect of H_2_S against H/R injury in rat hippocampal neurons (RHNs).

## 2. Results

### 2.1. His-ROCK_2_^wild^ Were Expressed in an Insoluble Form in E. Coli

To detect phosphorylation of ROCK_2_ at potential sites, prokaryotic plasmids, ROCK_2_^wild^-pET-32a(+), were constructed and transformed into *E. coli* to express histidine (His)-tagged ROCK_2_^wild^. The transformed *E. colis* were induced with 0.2 and 1.0 mmol/L isopropylbeta-D-thiogalactopyranoside (IPTG). As shown in Figure 1, the result of Coomassie blue staining indicated that a significantly different band appeared in the lysate of the transformed *E. colis* cultured at 15 °C overnight and the molecular weight of the band was about 170 KDa, as expected for His-ROCK_2_^wild^. After sonication of the lysate, the band was only present in the precipitation but not the supernatant of the transfected *E. colis*, preliminarily indicating that the His-ROCK_2_^wild^ was expressed in the transformed *E. colis* and that the protein may exist in the insoluble form and cannot be further purified. Western blot results showed that the molecular weight of the ROCK_2_ expressed in the ROCK_2_^wild^-pET-32a(+)-transformed *E. coli* was consistent with that in RHNs. Hence, the lysate precipitation of the *E. colis* was used for in vitro phosphorylation assays.

### 2.2. Effect of NaHS on the Related Sites’ Phosphorylation of His-ROCK_2_^wild^ In Vitro

By using an LC–MS/MS (liquid chromatography–tandem mass spectrometry) assay, the impact of NaHS, a donor of H_2_S, on the related sites’ phosphorylation of His-ROCK_2_^wild^ was investigated. As shown in Figure 2, in the absence of serine kinases PLK1 or threonine kinase PKA, no phosphorylation of any site of His-ROCK_2_^wild^ was observed and NaHS (100 μmol/L) had no effect on the phosphorylation of any site of His-ROCK_2_^wild^ in the absence of PLK1 or PKA. However, the addition of PLK1 (1 μg) and PKA (7.5 kU) led to the phosphorylation of GST-ROCK_2_^wild^ at the Ser575 and Ser1099 sites. The peptide spectrum match (PSM) score, the proteins identified by mass spectrometry, significantly increased to 2.67 ± 2.08 from 0.0 ± 0.0 in the Ser575 site (*p* < 0.01). The PSM score in the Ser1099 site changed from 2.0 ± 0 to 1.67 ± 0.58. In the presence of PLK1, NaHS abolished the Ser575 phosphorylation of GST-ROCK2wild (*p* < 0.01) but had no obvious effect on the increased PSM score of the Ser1099 phosphorylation (*p* > 0.05); in the presence of PKA, the phosphorylation of GST-ROCK_2_^wild^ at Thr436 occurred and NaHS remarkably decreased the PSM score of the Thr436 phosphorylation (*p* < 0.01). These results demonstrated that H_2_S could inhibit the phosphorylation of GST-ROCK_2_^wild^ at the Thr436 and Ser575 sites. 

### 2.3. Transfection with Recombinant Eukaryotic Plasmids by Lentivirusin in RHNs

In order to study the effect of NaHS on the phosphorylation of ROCK_2_ at Thr436 and Ser575 in RHNs, eukaryotic recombinant plasmids, ROCK_2_^wild^-pEGFP-N1, ROCK_2_^T436A^-pEGFP-N1, and ROCK_2_^S575F^-pEGFP-N1, were constructed by Gene Create Biological Engineering, Co. (Wuhan, China). The three types of eukaryotic recombinant plasmids and empty plasmid were transfected into RHNs using a lentiviral infection manner, respectively. The transfection efficiency of the plasmids was determined by observing green fluorescent protein (GFP) expression under an inverted microscope. As shown in Figure 3, compared to before transfection, GFPs were obviously expressed in RHNs at a transfection of 24, 48, 72, and 108 h. The GFP fluorescence intensity at a transfection of 72 h was significantly higher than that at other times, indicating that the transfection efficiency was the highest at 72 h. Therefore, the RHNs transfected for 72 h were selected for subsequent experiments. 

### 2.4. NaHS Inhibited the Expression of ROCK_2_ and GFP-ROCK_2_ in the Transfected RHNs

As shown in Figure 4, in the empty plasmid-transfected RHNs, ROCK_2_ was detected, and NaHS 50, 100, and 200 μmol/L significantly decreased the ROCK_2_ expression; in the eukaryotic recombinant plasmids-transfected RHNs, not only ROCK_2_ but also GFP-tagged ROCK_2_ including GFP-ROCK_2_^wild^, GFP-ROCK_2_^T436A^, and GFP-ROCK_2_^S575F^ were detected. NaHS 100 and 200 μmol/L markedly inhibited the ROCK_2_ and GFP-ROCK_2_^wild^ expressions in the ROCK_2_^wild^-pEGFP-N1-transfected RHNs and the ROCK_2_ and GFP-ROCK_2_^S575F^ expressions in the GFP-ROCK_2_^S575F^-pEGFP-N1-transfected RHNs; 200 μmol/L NaHS obviously inhibited the ROCK_2_ and GFP-ROCK_2_^T436A^ expression in the ROCK_2_^T436A^-pEGFP-N1-transfected RHNs. These results showed that H_2_S significantly inhibited the expressions of ROCK_2_ and the recombinant proteins GFP-ROCK_2_^wild^ and GFP-ROCK_2_^T436A^ as well as GFP-ROCK_2_^S575F^ in RHNs. 

### 2.5. Thr436- and Ser575-Mediated Inhibition of NaHS on ROCK_2_ Activity in RHNs

The effect of NaHS on ROCK_2_ activity in the transfected RHNs was further investigated using an ELISA assay. As shown in Figure 5A, compared with the empty plasmid-transfected RHNs, ROCK_2_ activity in the GFP-ROCK_2_^wild^-pEGFP-N1-transfected RHNs increased significantly (*p* < 0.05), but the ROCK_2_ activity in the GFP-ROCK_2_^T436A^-pEGFP-N1- or the GFP-ROCK_2_^S575F^-pEGFP-N1-transfected RHNs was obviously lower than that in the GFP-ROCK_2_^wild^-pEGFP-N1-transfected RHNs (*p* < 0.05). The result demonstrated that both the Thr436 and Ser 575 sites participate in the activation of ROCK_2_. Figure 5B,C showed that 50, 100, and 200 μmol/L of NaHS significantly decreased ROCK_2_ activity in the empty plasmid- or the ROCK_2_^wild^-pEGFP-N1-transfected RHNs (*p* < 0.05 or *p* < 0.01). However, 50 μmol/L of NaHS had no obvious effect on ROCK_2_ activity in the ROCK_2_^S575F^-pEGFP-N1-transfected RHNs (*p* > 0.05, Figure 5D). Even 100 μmol/L of NaHS did not affect ROCK_2_ activity in the ROCK_2_^T436A^-pEGFP-N1-transfected RHNs (*p* > 0.05, Figure 5E). The results indicated the Thr436 and Ser575 sites of ROCK_2_ may mediate the inhibitory effect of H_2_S on the ROCK_2_ activity in RHNs.

### 2.6. Role of Thr436 and Ser575 of ROCK_2_ in Protection of NaHS against H/R Injury in RHNs

Since the Thr436 and Ser575 sites may mediate the inhibition of H_2_S on ROCK_2_ activation in RHNs, and previous studies indicated that the inhibition of the RhoA-ROCK pathway could protect RHNs from H/R injury, the roles of Thr436 and Ser575 of ROCK_2_ in the protective effect of NaHS against H/R injury were investigated in the present study.

Figure 6 shows that, compared with the sham group, H/R exposure induced a significant injury, as indicated by the decreased cellular viability as well as the increased LDH and NSE levels in the supernatant in the four types of the transfected RHNs. As shown in Figure 6A(a),B(a),C(a), 100 and 200 μmol/L of NaHS could obviously inhibit the decreased cellular viability and the increased LDH and NSE in the empty plasmid-transfected RHNs. In the ROCK_2_^wild^-pEGFP-N1-transfected RHNs, not only 100 and 200 μmol/L of NaHS but also 50 μmol/L of NaHS markedly inhibited the decreased cellular viability and the increased LDH and NSE (Figure 6A(b),B(b),C(b)). However, in the ROCK_2_^T436A^-pEGFP-N1- or the ROCK_2_^S575F^-pEGFP-N1-transfected RHNs, 50 μmol/L of NaHS had no effect on the reduction in cellular viability and the increases in LDH and NSE; only 100 μmol/L or even 200 μmol/L of NaHS could obviously inhibit the decreased cell viability and the increased LDH and NSE (Figure 6A(c,d),B(c,d),C(c,d)). 

After obtaining the difference between the NaHS group and the H/R group and dividing by the value of the H/R group, we obtained the inhibition rate of different groups. In addition, Figure 7 shows that the inhibitions of 100 and 200 μmol/L of NaHS on the decreased cell viability and the increased LDH and NSE in the ROCK_2_^T436A^-pEGFP-N1- or the ROCK_2_^S575F^-pEGFP-N1-transfected RHNs were weaker than those in the ROCK_2_^wild^-pEGFP-N1-transfected RHNs. These results demonstrated that Thr436 and Ser575 of ROCK_2_ mediated the protective effect of H_2_S against an H/R injury in RHNs.

### 2.7. Role of Thr436 and Ser575 of ROCK2 in NaHS Reducing Intracellular Free Ca2+ in RHNs

As shown in Figure 8A,B, compared with the sham group, H/R injury significantly increased the fluorescence intensity of the intracellular free Ca^2+^ concentration ([Ca^2+^]_i_) in the four types of the transfected RHNs. As with the Ca^2+^ channel inhibitor nifedipine (Nif, 100 μM), 100 μmol/L of NaHS markedly inhibited the H/R injury-increased fluorescence intensity of [Ca^2+^]_i_ in each type of the transfected RHN. There was no significant difference among the inhibitions of Nif in the four types of the transfected RHNs. However, the inhibition of NaHS in the ROCK_2_^wild^-pEGFP-N1-transfected RHNs was significantly stronger than that in the empty plasmid-transfected RHNs. The inhibition of NaHS in either the ROCK_2_^T436A^-pEGFP-N1- or the ROCK_2_^S575F^-pEGFP-N1-transfected RHNs was remarkably weaker than that in the ROCK_2_^wild^-pEGFP-N1-transfected RHNs. The result indicated that Thr436 and Ser575 of ROCK_2_ are engaged with the inhibitory effect of H_2_S on [Ca^2+^]_i_ in RHNs.

## 3. Discussion

The present study focused on the roles of ROCK_2_ phosphorylation at related sites in the inhibitory effects of H_2_S on ROCK_2_ activation and against H/R injury in RHNs. The main findings include: (1) H_2_S inhibits the phosphorylation of ROCK_2_ at the Thr436 and Ser575 sites in vitro; (2) the Thr436 and Ser575 sites of ROCK_2_ mediate the inhibition of H_2_S on ROCK_2_ activation in RHNs; and (3) Thr436 and Ser575 of ROCK_2_ participate in the protective effect of H_2_S against an H/R injury via attenuating [Ca^2+^]_i_ in RHNs.

Phosphorylation refers to the transfer of phosphorylation of adenosine triphosphate (ATP) or GTP at the γ position to amino acid residues of substrate protein catalyzed by protein kinase. Phosphorylation regulation mainly occurs on serine, threonine, and tyrosine [22]. In the process of phosphorylation modification, the strongly negatively charged phosphate group is transferred to the specific amino acid of the protein, which will cause the overall configuration of the protein to change and the activity of the protein and the interaction between other molecules will also change. Phosphorylation plays different roles in the regulation of cellular physiological processes, such as signal transduction, gene expression, cell division, etc. [23,24,25]. Therefore, the identification and analysis of the phosphorylation of related sites in protein are helpful to find novel therapeutic targets for some diseases. In order to explore the effect of H_2_S on the phosphorylation of ROCK_2_ at some sites, prokaryotic plasmid ROCK_2_^wild^-pET-32a(+) was constructed and transfected into *E. colis* to express His-ROCK_2_^wild^ protein in the present study. By using Coomassie brilliant blue staining and Western blot assays, His-ROCK_2_^wild^ protein was detected in the lysate of the *E. colis* and the lysate precipitation but not in the supernatant, suggesting that His-ROCK_2_^wild^ protein exists in an insoluble form, which may be because ROCK_2_ is a macromolecular protein composed of more than 1300 amino acid residues. 

By using an LC–MS/MS assay, it was observed that, in the presence of serine kinase PLK1 or threonine kinase PKA, phosphorylation occurred at three sites, Thr436, Ser575 and Ser1099, in His-ROCK_2_^wild^ protein in vitro. However, the phosphorylation of these three sites did not take place in the absence of PLK1 or PKA, indicating that the phosphorylation of His-ROCK_2_^wild^ at the three sites cannot happen by autophosphorylation. As far as we know, the phosphorylation of ROCK_2_ at Ser1099 was reported previously [26], but the phosphorylation of ROCK_2_ at Thr436 or Ser575 was found for the first time in the present study. Furthermore, our study observed that the phosphorylation of His-ROCK_2_^wild^ at Thr436 or Ser575 was significantly inhibited by treatment with H_2_S donor NaHS, demonstrating that H_2_S could inhibit the phosphorylation of ROCK_2_ at both Thr436 and Ser575. 

The inhibition of H_2_S on the phosphorylation of His-ROCK_2_^wild^ at Thr436 or Ser575 seems contrary to our recent report on the promotion of H_2_S on the phosphorylation of ROCK_2_ at Tyr722 [27]. It was already known that the phosphorylation of different sites of ROCK_2_ had different effects on its activity; the phosphorylation of some sites can enhance the activity of ROCK_2_, but the phosphorylation of other sites can inhibit its activity. This study used eukaryotic recombinant ROCK_2_ wild-type plasmid (ROCK_2_^wild^-pEGFP-N1) and plasmids mutated at Thr436 and Ser575 (ROCK_2_^T436A^-pEGFP-N1 and ROCK_2_^S575F^-pEGFP-N1) to transfect RHNs for the detection of ROCK_2_ expression and activity. In the empty plasmid-transfected RHNs, only ROCK_2_ was examined. Nevertheless, in the ROCK_2_^wild^-pEGFP-N1-, the ROCK_2_^T436A^-pEGFP-N1-, or the ROCK_2_^S575F^-pEGFP-N1-transfected RHNs, in addition to ROCK_2_, GFP-ROCK_2_^wild,^ GFP-ROCK_2_^T436A,^ and GFP-ROCK_2_^S575F^ were detected, respectively, confirming that the transfection was successful. Our result showed that NaHS significantly inhibited ROCK_2_ protein expression in each type of plasmid-transfected RHN. This is consistent with a previous study that NaHS inhibits cerebral I/R injury-increased ROCK_2_ protein expression in mouse hippocampus [28]. Our results also indicated that NaHS not only attenuated GFP-ROCK_2_^wild^ protein expression but also decreased GFP-ROCK_2_^T436A^ and GFP-ROCK_2_^S575F^ protein expressions, indicating that both Thr436 and Ser575 sites did not take part in the inhibitory effect of H_2_S on ROCK_2_ expression in RHNs. However, the present study revealed that both Thr436 and Ser575 participate in the activation of ROCK_2_ and mediate the inhibitory effect of H_2_S on ROCK_2_ activity in RHNs. Combining with the aforementioned result of NaHS inhibiting the phosphorylation of ROCK_2_ at Thr436 and Ser575 in vitro, our study demonstrated that H_2_S could inhibit the activation of ROCK_2_ via attenuating the phosphorylation of ROCK_2_ at Thr436 and Ser575. This is not contradictory to its inhibition on ROCK_2_ activity by promoting the phosphorylation of ROCK_2_ at Tyr722. 

The over-activation of ROCK_2_ is involved in the pathological process of cerebral stroke. It was reported that KD-025, a ROCK_2_ selective inhibitor, has a good efficacy on cerebral ischemic injury in mice [29]. Therefore, the role of H_2_S regulating the phosphorylation of ROCK_2_ at Thr436 and Ser575 in its neuroprotection against H/R injury was investigated in this study. LDH is an important intracellular enzyme in glycolysis, with a tetrameric structure, which can catalyze the conversion of pyruvate to lactic acid and oxidize nicotinamide adenine dinucleotide dehydrogenase (NADH) to nicotinamide adenine dinucleotide (NAD+) [30]. NSE is also an intracellular enzyme, highly specific for nerve cells, and can be used to quantitatively detect brain injury and provide improved diagnosis and outcome evaluation for patients with ischemic stroke, cerebral hemorrhage, seizures, cardiac arrest, coma after cardiopulmonary resuscitation, and traumatic brain injury [31]. When nerve cells are injured, LDH and NSE are released into the cellular environment. Thus, the release of both LDH and NSE is a reliable index of cerebral damage detection. In the present study, H/R injury in RHNs was indicated by a significant decrease in cellular viability and an obvious release of LDH and NSE (increase in supernatant LDH and NSE). Unlike the Ca^2+^ channel blocker Nif, NaHS had an obvious difference in protecting against H/R injury in the empty plasmid-, the ROCK_2_^wild^-pEGFP-N1-, the ROCK_2_^T436A^-pEGFP-N1-, and the ROCK_2_^S575F^-pEGFP-N1-transfected RHNs. The results showed that the protection of NaHS in the ROCK_2_^wild^-pEGFP-N1-transfected RHNs was more patent than that in the empty plasmid-, the ROCK_2_^T436A^-pEGFP-N1-, and the ROCK_2_^S575F^-pEGFP-N1-transfected RHNs, demonstrating that Thr436 and Ser575 of ROCK_2_ mediate the neuroprotection of H_2_S against an H/R injury in RHNs. Therefore, it was possible to conclude that H_2_S attenuates the phosphorylation of ROCK_2_ at Thr436 and Ser575 to inhibit ROCK activation so as to protect RHNs from H/R injury. 

The [Ca^2+^]_i_ is a key factor with a regulatory role in a variety of cell functions. Intracellular [Ca^2+^]_i_ overload plays a major role in the cerebral I/R process [32]. It was reported that an H/R-induced increase in [Ca^2+^]_i_ was significantly inhibited by treatment with the ROCK inhibitor fasudil or Y-27632 [33]. In the present study, the change in [Ca^2+^]_i_ in each type of the transfected RHNs in the absence and presence of NaHS was detected. The result indicated that Thr436 and Ser575 of ROCK_2_ are engaged with the inhibitory effect of H_2_S on H/R-increased [Ca^2+^]_i_ in RHNs. Taken together with the result of the inhibition of H_2_S on the activation of ROCK_2_, our results suggested that the inhibition of H_2_S on the increased [Ca^2+^]_i_ may be owed to its attenuation on the phosphorylation of ROCK_2_ at Thr436 and Ser575. This may be related to the protection of H_2_S against H/R injury in RHNs.

However, this experiment still had some limitations. For example, we did not obtain phosphorylated antibodies for p-ROCK_2_(Thr436) and p-ROCK_2_ (Ser575) so that the phosphorylation of ROCK_2_ in the cell model could be observed. Meanwhile, after we changed the two labels, the ROCK_2_ was still insoluble; other expression systems may exist to promote the solubility of the protein. Based on the interesting findings, we intend to further explore whether these sites can mediate the protective effect of H_2_S against I/R injury in animal models. 

## 4. Materials and Methods

### 4.1. Reagents

NaHS was obtained from Sigma Chemical (St. Louis, MO, USA). Tris-Base was purchased from Amersco (St. Louis, MO, USA). Ethanol, glacial acetic acid, Na_2_S_2_O_3_, sodium acetate, glutaraldehyde, silver nitrate, formaldehyde, sodium carbonate, and dithiothreitol (DTT) were purchased from Sinopharm Chemical Reagent Co., Ltd. (Shanghai, China). Cell counting kit-8, ATP, PLK1, PKA, and IPTG were purchased from Abcam (San Francisco, CA, USA). The LDH assay kit was purchased from Servicebio (Wuhan, China). The NSE assay kit was obtained from Jiangsu Meimian Industrial, Co., Ltd. (Jiangsu, China). Flou-8 AM was purchased from biofount (Beijing, China).

### 4.2. Plasmids and Bacterium

We commissioned Gene Create Biological Engineering, Co. (Wuhan, China) to construct the prokaryotic plasmids ROCK_2_^wild^-pET-32a(+) and eukaryotic recombinant plasmids ROCK_2_^wild^-pEGFP-N1, ROCK_2_^T436A^-pEGFP-N1, and ROCK_2_^S575F^-pEGFP-N1. *E. coli* (BL21) was obtained from Gene Create Biological Engineering, Co. (Wuhan, China). ROCK_2_^T436A^ means Thr436 was mutated to an alanine (Ala or A). ROCK_2_^S575F^ means Ser575 was mutated to a phenylalanine (Phe or F). All the mutations were at the mRNA level. The sequencing results showed that the plasmids and the mutations were successfully constructed. (See Appendix A.)

### 4.3. Expression of the Prokaryotic Recombinant Proteins His-ROCK_2_^wild^

A total of 10 μL of glycerol bacteria containing the prokaryotic plasmid ROCK_2_^wild^-pGEX-6P-1 was added to the solid Luria-Bertani (LB) medium. These were spread evenly and quickly by using a coating rod. The coated dishes were inverted overnight in an incubator at 37 °C. Single colonies with a good growth status using a pipette gun were inoculated into a centrifuge tube containing 20 mL of liquid LB medium in a shaker (37 °C, 230 rpm). Using medium as a blank control and looking at the OD solution of about 0.7, IPTG-induced expression protein was added at final concentrations of 0.2 mM and 1.0 mM, respectively. The fluids supplemented with different final concentrations of IPTG were induced overnight or for 4 h at 15 °C and 37 °C. The solution was collected at 10,000 rpm for 10 min and concentrated by adding 2 mL of PBS. The concentrated bacterial solution was crushed for 30 min using an ultrasonic crusher (100 W). The crushed bacterial solution was centrifuged at 10,000 rpm for 10 min. An equal volume of PBS solution was added into the precipitate and the supernatant. Some of the bacterial solution after ultrasonic crushing was sent to Shanghai Sangong Biological Co., Ltd. (Shanghai, China) for base sequencing. An upper buffer was added to the supernatant and precipitated samples; we denatured the protein at 100 °C. Electrophoresis was performed using an 8% SDS-PAGE gel for 40 min, with an electrophoresis meter voltage set to 120 V. After the electrophoresis, the gel was removed and the protein expression was visualized by staining using a Coomassie blue rapid staining solution for 10 min in a shaker. Sequencing of the eukaryotic plasmids ROCK2T436A-pEGFP-N1 showed that ACA (Thr) was mutated into GCG (Ala) at 436 and TCT (Ser) was mutated into TTT (Phe) at 575 of ROCK_2_^S575F^-pEGFP-N1 (see Appendix A). 

### 4.4. Culture of Primary RHNS

The hippocampus was carefully removed with eye tweezers. Peripheral blood vessels were removed as far as possible and transferred to a solution containing 0.125% trypsin (0.25% trypsin:PBS = 1:1). The tissue was completely cut up and put into a 37 °C constant temperature cell incubator for digestion. The centrifuge tube containing the tissue was shaken for 20 mins. The preheated DMEM medium was added after digestion and then filtered after blowing evenly. The solution was centrifuged at 1400 rpm for 5 min. The supernatant was poured off, the complete medium was added, and the cells were gently blown and resuspended. After a cell count, the cells were inoculated into the 24-well plate (poly-DL-lysine coated) and cultured in a constant temperature cell incubator. After 24 h, we performed half-dose fluid replacement with Neurobasal medium and performed full fluid replacement every 1 to 2 days for incubation for 5 to 6 days.

### 4.5. Phosphorylation Test In Vitro

A total of 15 μL of lysate of the ROCK_2_^wild^-pGEX-6P-1-transfected *E. colis* induced by IPTG and 10 μL of 100 μmol/L ATP was added to 30 μL of 10×Kinase buffer (20 mM Tris-Cl, 100 mM KCl, 2 mM EGTA, 5 mM MgCl_2_, pH 7.4) without or with 3 μL of kinases PLK1 0.52 mg/mL (or PKA 2500 KU/mL) and mixed fully. The mixture was shaken at 180 rpm under 37 °C for 30 min. The phosphorylation reaction was terminated by adding 7 μL of 5× loading buffer (250 mM Tris-HCl, 10% SDS, 0.5% BPB, 50% glycerol, 5% β-mercaptoethanol). The mixture was kept at 100 °C for 10 min to denature protein. A small amount of the mixture was loaded onto an 8% SDS-PAGE gel to perform Western blot assay for detecting His-ROCK_2_^wild^ protein expression; the rest was also loaded onto the 8% SDS-PAGE gel for electrophoresis and then stained with the Coomassie brilliant blue. Using the phosphorylation reaction mixture of lysates of *E. coli* induced without IPTG as a control, the differential protein band in the Coomassie bright blue-stained SDS-PAGE gel was determined; this should have had the same position as the His-ROCK band in the Western blot assay. A differential protein band was excised and cut into patches of approximately 1 mm × 1 mm. The patches were cleaned with a glue cleaning solution, dehydrated using acetonitrile, and then sent to Hooper Biotechnology Co., Ltd. (Shanghai, China) for LC–MS/MS assay to detect the phosphorylation of ROCK_2_ at related sites.

### 4.6. LC–MS/MS

A mixed patch with 8 M of UA in the 10 K filter unit was centrifuged at 14,000× *g* for 15 min. We added 200 μL of UA and centrifuged at 14,000 g for 15 min. We discarded the flow-through from the collection tube. We alkylated the proteins with IAA and incubated them for 45 min in the dark. We discarded the flow-through. We added 100 μL of UA and centrifuged at 14,000× *g* for 15 min; this step was repeated once. We added 200 μL of 50 mM ABC and centrifuged at 14,000× *g* for 15 min; this step was repeated once. We changed to a new collection tube, added 4 μg of trypsin, and incubated at 37 °C for 16 h. We centrifuged this at 14,000× *g* for 10 min and collected the flow-through in a new tube. We added 50 mM of ABC, centrifuged at 14,000× *g* for 10 min, collected the flow-through to the above tube, and dried the sample with SpeedVac. The sample was dissolved in 0.1% TFA, desalted with C18 ZipTips, and dried with SpeedVac. The sample was resuspended with 0.1% formic acid for mass spectrometry analysis. The peptide samples were analyzed on a Thermo Fisher LTQ Obitrap ETD mass spectrometry. Briefly, we loaded the sample onto an HPLC chromatography system called a Thermo Fisher Easy-nLC 1000 equipped with a C18 column (1.8 mm, 0.15 × 100 mm). Solvent A contained 0.1% formic acid and solvent B contained 100% acetonitrile. The elution gradient was from 4% to 18% in 182 min and 18% to 90% in 13 min for solvent B at a flow rate of 300 nL/min. Mass spectrometry analyses were carried out at AIMSMASS Co., Ltd. (Shanghai, China) in the positive-ion mode with an automated data-dependent MS/MS analysis with full scans (350–1600 m/z) acquired using FTMS at a mass resolution of 30,000; the 10 most intense precursor ions were selected for MS/MS. The MS/MS was acquired using higher-energy collision dissociation at 35% collision energy at a mass resolution of 15,000.

### 4.7. Lentiviral Transfection

The lentivirus vector was a second-generation vector that was cotransfected into 293T cells by empty, ROCK_2_^wild^-pEGFP-N1, ROCK_2_^T436A^-pEGFP-N1, and ROCK_2_^S575F^-pEGFP-N1 plasmids; lentivirus packaging plasmid pCD/NLBH*DDD; and membrane protein expression plasmid PLTR-G. RHNs in a good growth state were selected and centrifuged at 10,000 rpm for 10 min. Then, the cells were resuspended with a fresh cell culture medium and counted. The suspension of RHNs (5 × 10^5^/mL) was inoculated into 24 wells of a medium; the lentivirus expressing ROCK_2_^wild^, ROCK_2_^T436A,^ and ROCK_2_^S575F^ (1 × 108 uA/mL) was added to the well plate when the degree of cell fusion reached 70%. After 24 h of culture, the medium was replaced with a fresh medium. The expression of GFP protein at 24 h, 48 h, 72 h, and 108 h was respectively observed under an inverted fluorescence microscope to evaluate the transfection efficiency.

### 4.8. Western Blot

The 8% SDS-PAGE (7.5 mL top glue premix, 7.5 mL bottom glue premix, 2 μL tetramethylethylene-diamine) was prepared according to the molecular weight of ROCK_2_. Then, 15 μL of cell lysate was added to each well. After electrophoresis for 180 min at 120 V, the target protein and GAPDH were transferred to a PVDF membrane using wet membrane transfer, blocked at room temperature using 5% skim milk for 1 h, and then transferred to TBST buffer and washed twice for 10 min each. The membranes were individually incubated overnight in the anti-ROCK_2_ antibody solution for 4 °C and then incubated in the secondary antibody solution for 1 h at room temperature. The ECL-plus development solution was prepared and applied evenly on the PVDF membrane. The chemiluminescence was visualized using a Fluor-S-max imager. Gray values of the different bands were analyzed using Image J software; the ratios of the ROCK_2_ and GAPDH were calculated and counted using Graphpad prism 8.0 software.

### 4.9. Determination of ROCK2 Activity in RHNs

According to the manufacturer’s instructions, ROCK_2_ activity in RHNs was measured. The prepared RHNs were crushed with a 100 W ultrasonic crusher and centrifuged at 10,000 rpm for 10 min. The supernatant was added to the 96-well plate and incubated at 37 °C for 30 min. Then, 50 µL of the PBS was added to each well, followed by 50 µL of chromogenic agent A and chromogenic agent B each. The mixture was gently shaken, mixed well, and incubated at 37 °C for 15 min in the dark. Then, 50 µL of the PBS was added to each well to terminate the reaction. The activity of ROCK_2_ was measured at a wavelength of 450 nm using a microplate reader.

### 4.10. Establishment of H/R Injury

RHNs were incubated in a hypoxia incubator (1% O_2_, 95% N_2_, 4% CO_2_) at 37 °C for 4 h and then cultured at 37 °C under normoxic conditions (37 °C, 95% O_2_, 5% CO_2_) for 12 h. The RHNs in the control group were kept under normoxic conditions without hypoxia [18]. 

### 4.11. Determination of Cell Viability

Cell viability was determined using a cell counting kit (CCK-8). Briefly, the RHNs’ suspension was transferred into a 96-well plate, and a total of 10 µL of the CCK-8 solution was added to each well. Then, the RHNs were incubated in an incubator (37 °C, 5% CO_2_) for 24 h. The absorbance at 450 nm was measured with a microplate reader.

### 4.12. Determination of the LDH and NSE Activities

LDH and NSE levels in the RHNs’ culture supernatant were respectively determined using the commercial assay kits. Briefly, the prepared RHNs’ suspension was centrifuged at 10,000 rpm for 10 min using a centrifuge (Eppendorf, Germany); a microplate reader was used to respectively detect LDH and NSE activities in the supernatant at 450 nm, according to the protocol of the LDH and NSE assay kits. 

### 4.13. Determination of [Ca^2+^]_i_

RHNs were incubated with a final concentration of 10 μM of Fluo-8 AM for 10 min at room temperature and then washed three times with PBS buffer. The [Ca^2+^]_i_ fluorescence intensity in the RHNs was measured using a Ca^2+^ imaging system fluorescence microscope.

Mature RHNs cultured for 6–8 days were selected, incubated with 10 μM of Fluo-8 AM for 10 min at room temperature, washed three times with PBS buffer, and preincubated with a normal physiological saline solution (NPSS) that contained 140 mM of NaCl, 10 mM of glucose, 5 mM of KCl, 5 mM of Hepes, 1 mM of CaCl_2,_ and 1 mM MgCl_2;_ the fluorescence intensity of Ca^2+^ in the different groups of RHNs was observed using a Ca^2+^ imaging system fluorescence microscope (excitation wavelength: 488 nm, emission wavelength: 514 nm).

### 4.14. Statistical Analysis

All data are represented as the mean ± SD. The t-test was used to compare the differences between the two groups. One-way analysis of variance (one-way ANOVA) and two-way ANOVA were used for multi-group comparisons. A *p* < 0.05 was considered as a statistically significant difference.

## 5. Conclusions

The present study is the first attempt to explore the regulation of H_2_S on the phosphorylation of ROCK_2_ at potential sites except Tyr722 and the role of this regulation in H_2_S inhibiting ROCK_2_ activation and protecting RHNs from H/R injury. The present study demonstrated that H_2_S attenuates the H/R injury-induced ROCK activation to decrease [Ca^2+^]_i_ via inhibiting the phosphorylation of ROCK_2_ at Thr436 and Ser575 and subsequently protects RHNs from H/R injury. Our findings not only contribute to a deeper understanding on the mechanism of H_2_S against cerebral I/R injury but also provide potential targets for the treatment of ischemic cerebral stroke.

## Figures and Tables

**Figure 1 pharmaceuticals-16-00218-f001:**
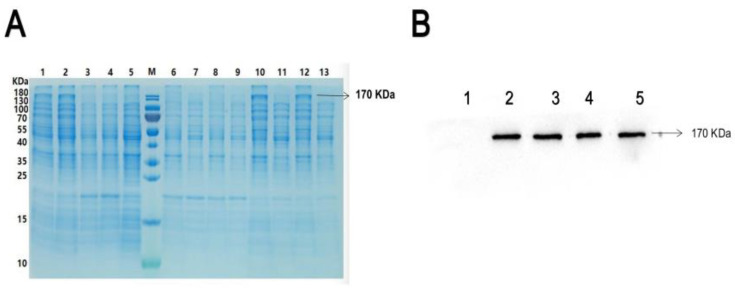
ROCK_2_^wild^ expression in the ROCK_2_^wild^-pET-32a(+)-transformed *E. colis* and RHNs. (**A**) Expression of His-ROCK_2_^wild^ by Coomassie brilliant blue. Marker (M) is the transformed *E. colis* cultured at 15 °C overnight with IPTG 0.2 mmol/L (1), at 15 °C overnight with IPTG 1.0 mmol/L (2), at 37 °C for 4 h with IPTG 0.2 mmol/L (3), at 37 °C for 4 h with IPTG 1.0 mmol/L (4), and at 15 °C overnight without IPTG (5); precipitation (6) and supernatant (7) of the transformed *E. colis* cultured at 37 °C with IPTG 1.0 mmol/L; precipitation (8) and supernatant (9) of the transformed *E. colis* cultured at 37 °C with IPTG 0.2 mmol/L; precipitation (10) and supernatant (11) of the transformed *E. colis* cultured at 15 °C with IPTG 1.0 mmol/L; precipitation (12) and supernatant (13) of the transformed *E. colis* cultured at 15 °C with IPTG 0.2 mmol/L. (**B**) Expression of ROCK_2_ by Western blot assay. (1) Empty plasmid-transformed *E. colis*; (2) lysate of RHNs; (3) supernatant of RHNs’ lysate; (4) lysate of ROCK_2_^wild^-pET-32a(+)-transformed *E. colis*; (5) precipitation of ROCK_2_^wild^-pET-32a(+)-transformed *E. colis*.

**Figure 2 pharmaceuticals-16-00218-f002:**
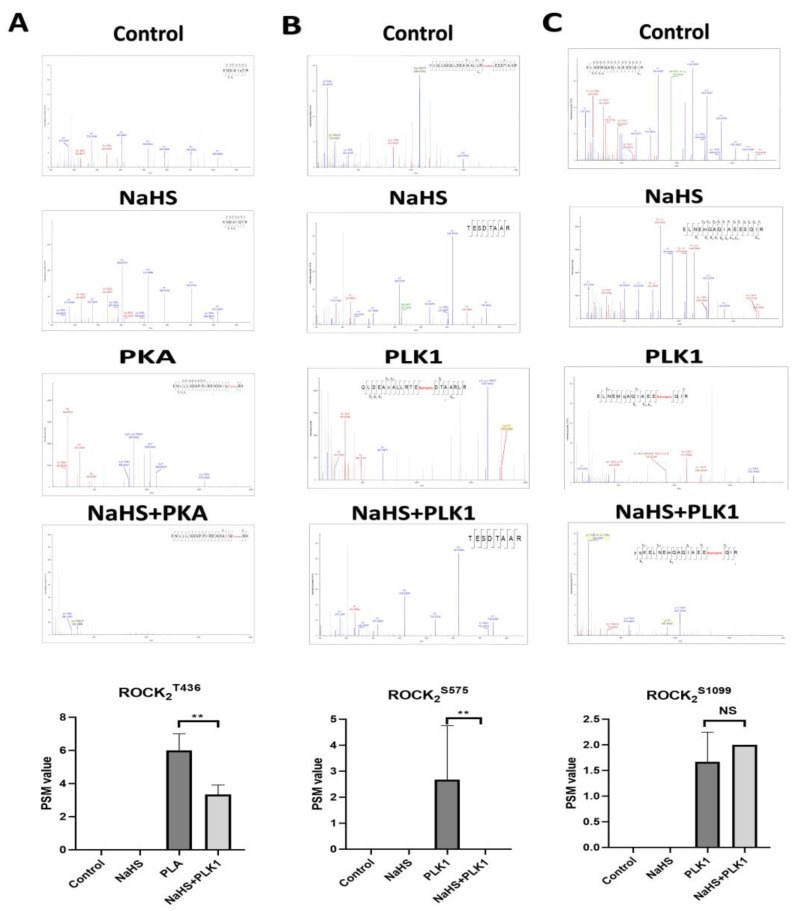
Effect of NaHS on phosphorylation of His-ROCK_2_^wild^ at Thr436 and Ser575 in vitro (LC–MS/MS assay, mean ± SD, n = 3). (**A**) Phosphorylation at Thr436. Mass spectrometry analysis images indicated: ROCK_2_ 430–438: ENDAIQTR in the control group and the NaHS group, ENDAIQT (phospho)RK in the PKA group. (**B**) Phosphorylation at Ser575. Mass spectrometry analysis images indicated: ROCK_2_ 573–580: TESDTAAR in the control group and the NaHS group, TES(phospho)DTAAR in the PLK1 group. (**C**) Phosphorylation at Ser1099. Mass spectrometry analysis images indicated: ROCK_2_ 1087–1102: ELNEMQAQIAEESQIR in the control group and the NaHS group, ELNEMQAQIAEES(phospho)QIR in the PLK1 group. ** *p* < 0.01 vs. the control group.

**Figure 3 pharmaceuticals-16-00218-f003:**
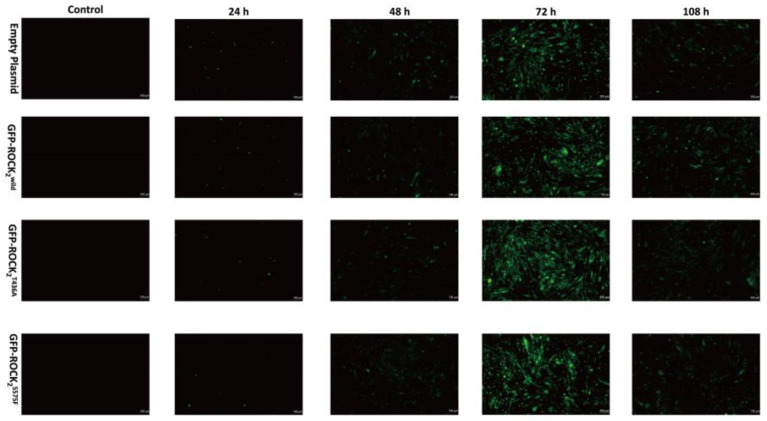
Representative images of RHNs transfected with empty, ROCK_2_^wild^-pEGFP-N1, ROCK_2_^T436A^-pEGFP-N1, and ROCK_2_^S575F^-pEGFP-N1 plasmids (100 μm). Transfection efficiency of the plasmids was determined by expression of green fluorescent protein (GFP). GFP showed green fluorescence.

**Figure 4 pharmaceuticals-16-00218-f004:**
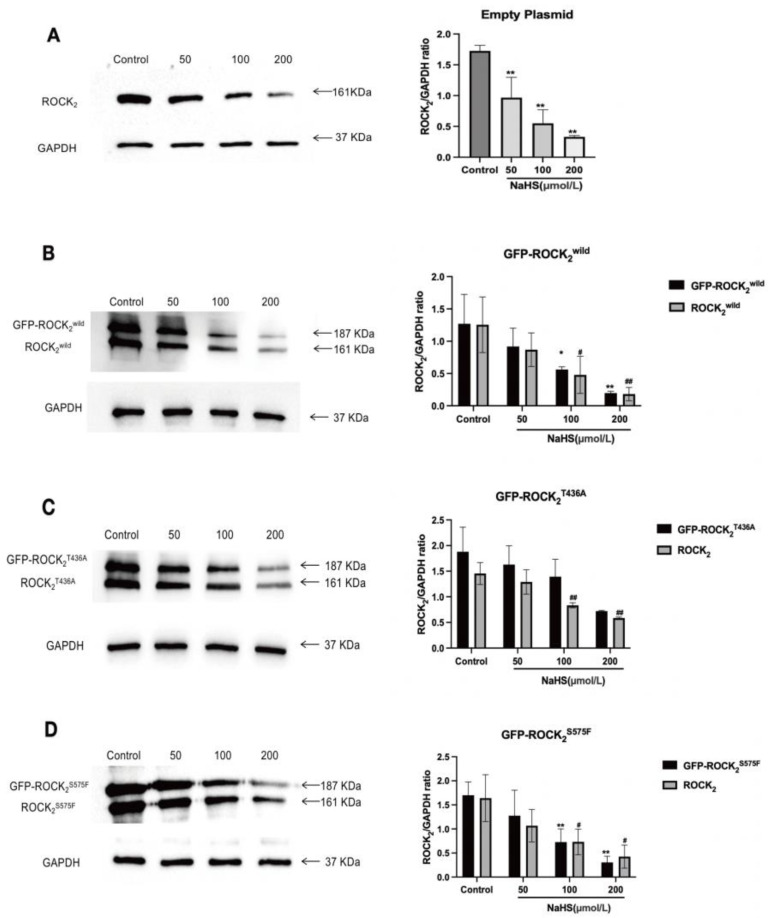
Effects of NaHS on ROCK_2_, GFP-ROCK_2_^wild^, GFP-ROCK_2_^T436A,^ and GFP-ROCK_2_^S575F^ expressions in the transfected RHNs. (**A**) Empty plasmid-transfected RHNs. ** *p* < 0.01 vs. control group. (**B**) GFP-ROCK_2_^wild^-transfected RHNs. * *p* < 0.05, ** *p* < 0.01 vs. control group (GFP-ROCK_2_^wild^), ^#^
*p* < 0.05, ^##^
*p* < 0.01 vs. control group (ROCK_2_^wild^). (**C**) GFP-ROCK_2_^T436A^-transfected RHNs. ** *p* < 0.01 vs. control group (GFP-ROCK_2_^T436A^), ^##^
*p* < 0.01 vs. control group (ROCK_2_^T436A^). (**D**) GFP-ROCK_2_^S575F^-transfected RHNs. ** *p* < 0.01 vs. control group (GFP-ROCK_2_^S575F^), ^#^
*p* < 0.05 vs. control group (ROCK_2_^S575F^). (Mean ± SD, n = 3).

**Figure 5 pharmaceuticals-16-00218-f005:**
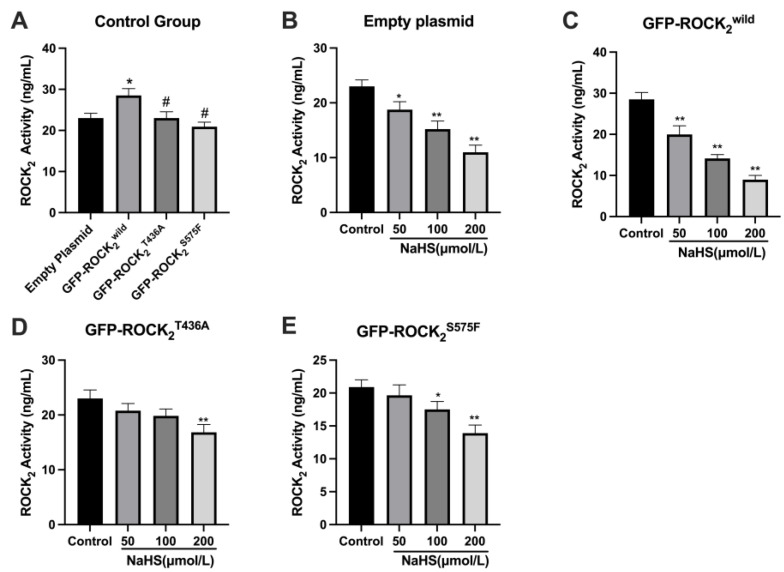
Thr436 and Ser575 mediated the inhibition of NaHS on ROCK_2_ activity in transfected RHNs. (**A**) Comparison of ROCK_2_ activity of the control group. * *p* < 0.05 vs. empty plasmid group, ^#^
*p* < 0.01 vs. GFP-ROCK_2_^wild^ group. (**B**) Empty plasmid-transfected RHNs. * *p* < 0.05, ** *p* < 0.01 vs. control group. (**C**) GFP-ROCK_2_^wild^-transfected RHNs. ** *p* < 0.01 vs. control group (GFP-ROCK_2_^wild^). (**D**) GFP-ROCK_2_^S575F^-transfected RHNs. ** *p* < 0.01 vs. control group (GFP-ROCK_2_^S575F^). (**E**) GFP-ROCK_2_^T436A^-transfected RHNs. * *p* < 0.05, ** *p* < 0.01 vs. control group (GFP-ROCK_2_^T436A^). (Mean ± SD, n = 3.)

**Figure 6 pharmaceuticals-16-00218-f006:**
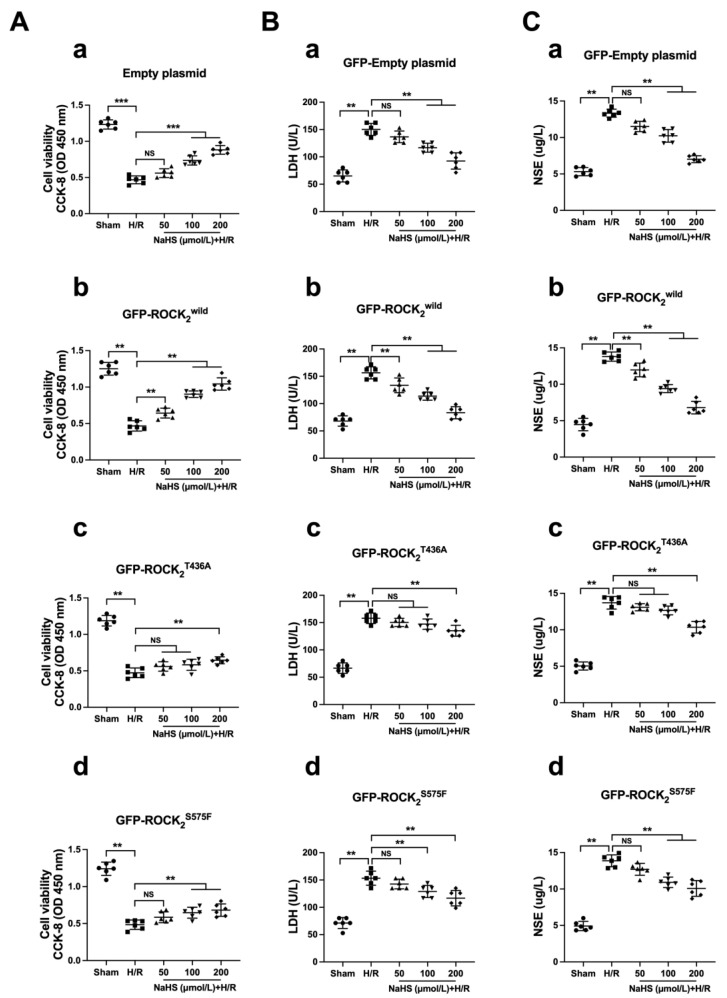
Thr436 and Ser575 mediate the effect of NaHS against H/R injury. (**A**) Cell viability: (**a**) empty plasmid group; (**b**) GFP-ROCK_2_^wild^ group; (**c**) GFP-ROCK_2_^T436A^; (**d**) GFP-ROCK_2_^S575F^. (**B**) Release of LDH: (**a**) empty plasmid group; (**b**) GFP-ROCK_2_^wild^ group; (**c**) GFP-ROCK_2_^T436A^; (**d**) GFP-ROCK_2_^S575F^. (**C**) Release of NSE: (**a**) empty plasmid group; (**b**) GFP-ROCK_2_^wild^ group; (**c**) GFP-ROCK_2_^T436A^; (**d**) GFP-ROCK_2_^S575F^. ** *p* < 0.01, *** *p* < 0.001. (Mean ± SD, n = 6.)

**Figure 7 pharmaceuticals-16-00218-f007:**
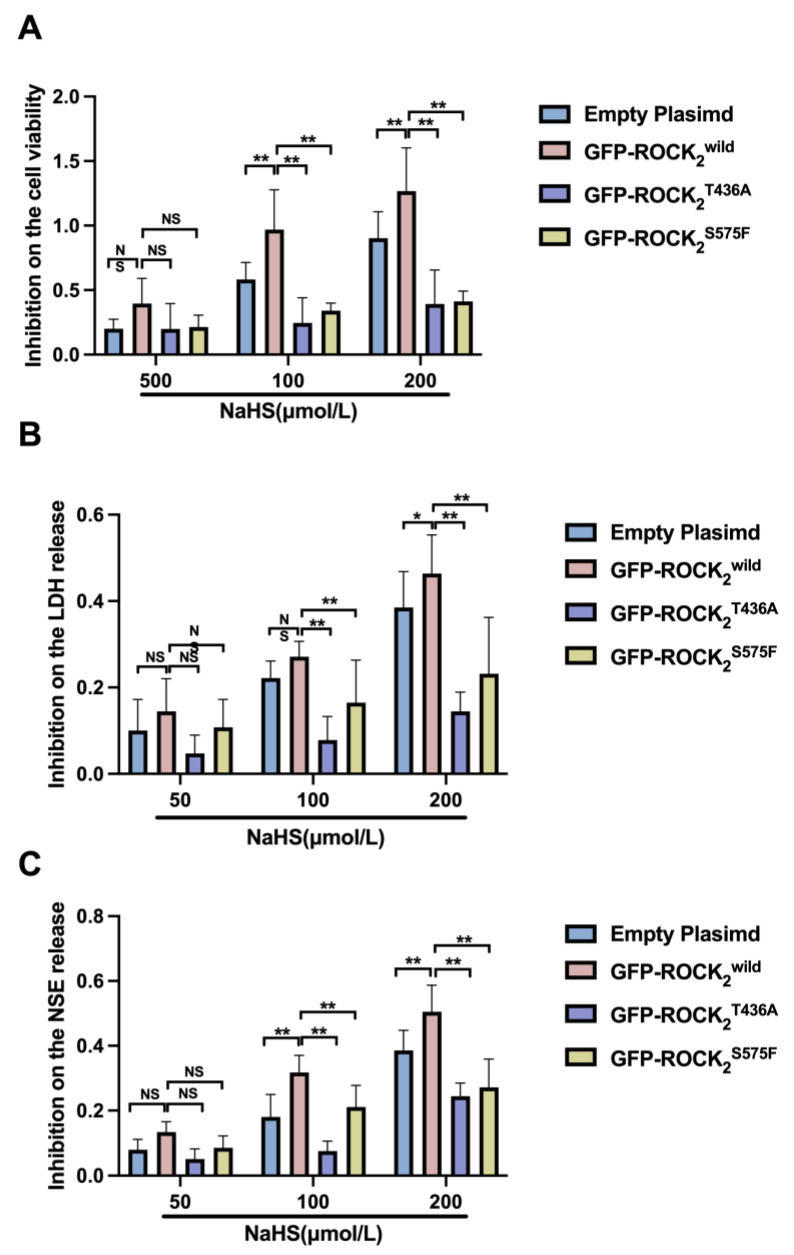
Thr436 and Ser575 mediated the inhibitory effect of NaHS on cell viability reduction and release of LDH and NSE. Comparison of inhibition rate of (**A**) cell viability reduction and (**B**) LDH and (**C**) NSE release. * *p* < 0.05, ** *p* < 0.01. (Mean ± SD, n = 6.)

**Figure 8 pharmaceuticals-16-00218-f008:**
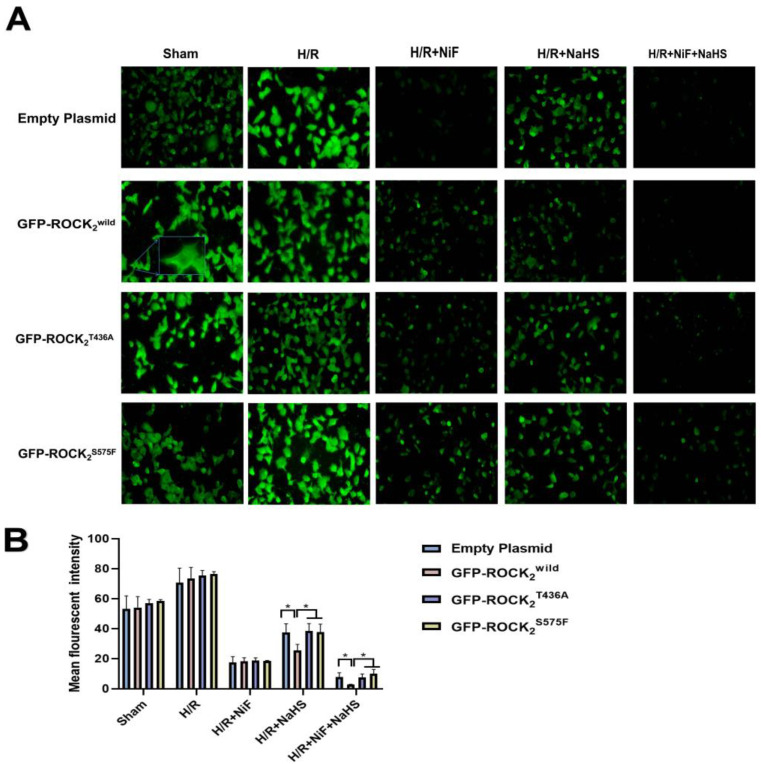
Thr436 and Ser575 mediated the inhibitory effect of NaHS on the fluorescence intensity of Ca^2+^. (**A**) Representative fluorescence images of Ca^2+^ (100 μm). (**B**) Comparison of the fluorescence intensity of Ca^2+^. * *p* < 0.05. (Mean ± SD, n = 3.)

## Data Availability

Data is contained within the article and Appendix A.

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
