# Peer review of "Protection of H2S against Hypoxia/Reoxygenation Injury in Rat Hippocampal Neurons through Inhibiting Phosphorylation of ROCK2 at Thr436 and Ser575"

_pharmaceuticals, 2023, doi:10.3390/ph16020218_

Round 1
Reviewer 1 Report
This study was conducted on the protection of H2S against hypoxia/reoxygenation injury in rat hippocampal neurons through inhibiting phosphorylation of 3 ROCK2 at Thr436 and Ser575. I found it interesting while going through this manuscript. However, the authors need to clarify the following concerns raised by me-
Line 48: should be pharmacological.
Line 54: Need Space after “brain”. Similar errors are found and please correct them throughout the manuscript.
Line 73: should be nerve cells.
Line: 83-85: Delete these statements.
Line 102: spacing error of “transferred”, please make sure to the consistent and correct use of this word.
Line 104, 105: should be “overnight”.
Line 128: write as Thr436 and Ser575 sites.
Line 144: I would suggest adding the results of transfected cells in percentage along with gross observation.
Line 170: The asterisk should be in superscript form and please check throughout.
Line 174: I recommend deleting the words “Western blot assay”. Because every experiment should be conducted in triplicate measurements. Correct them accordingly in other places.
Line 190: In Figure 5, the font size of the titles is not homogenous, correct them.
Line 192: Why do authors use a colon right after parentheses? I found the same in all figure legends.
Line 201: Give a citation after the statement.
Line 222: In figure 6, please increase the font size of the titles of each figures. It is not legible to read.
Line 244: In figure 8, adjust the figure position, there are many empty places between figures. Correct the title direction of the figures. Please give in the box of any corner one figure with an enlarged single neuron, so that it would be interesting to see the Ca positive puncta clearly.
Line 330: In the last paragraph of the discussion section, please add the limitations of your study and what could be your future plan for translational research, based on your present findings.
Line 388-389: Rewrite the sentence.
Line 390-391: Rephrase the sentence.
Line 393: Was it a Poly-DL-Lysine coated plate? Mention there.
Line 477: In H/R injury procedure, add a reference.
Line 483: Which cell counting kit did you use? Add details of company information.
Line 488: Which commercial company assay kits did you perform? Again lacking information, please add them.
Line 490: Add the model and company information of the centrifugal machine.
Line 496: Add company information of Fluo-8 AM.
Author Response
Line 48: should be pharmacological.
Answer: Thank you very much for your helpful comment. We have corrected the error in the revised manuscript.
Line 54: Need Space after “brain”. Similar errors are found and please correct them throughout the manuscript.
Answer: Thank you very much for your helpful comment. We have corrected the errors in the revised manuscript.
Line 73: should be nerve cells.
Answer: Thank you very much for your helpful comment. We have corrected the error in the revised manuscript.
Line: 83-85: Delete these statements.
Answer: Thank you very much for your helpful comment. We have deleted those statements in the revised manuscript.
Line 102: spacing error of “transferred”, please make sure to the consistent and correct use of this word.
Answer: Thank you very much for your helpful comment. We have corrected the errors in the revised manuscript.
Line 104, 105: should be “overnight”.
Answer: Thank you very much for your helpful comment. We have corrected the errors in the revised manuscript.
Line 128: write as Thr436 and Ser575 sites.
Answer: Thank you very much for your helpful comment. We have corrected the error in the revised manuscript.
Line 144: I would suggest adding the results of transfected cells in percentage along with gross observation.
Answer: Thank you very much for your helpful comment. We are very sorry that we didn't leave pictures to calculate the percentage, because the number of cell planks during transfection is the same. There was no significant difference in the number of cells during observation, but the transfection effect was significantly different.
Line 170: The asterisk should be in superscript form and please check throughout.
Answer: Thank you very much for your helpful comment. We have corrected the errors in the revised manuscript.
Line 174: I recommend deleting the words “Western blot assay”. Because every experiment should be conducted in triplicate measurements. Correct them accordingly in other places.
Answer: Thank you very much for your helpful comment. We have deleted the words “Western blot assay” and corrected others in the revised manuscript.
Line 190: In Figure 5, the font size of the titles is not homogenous, correct them.
Answer: Thank you very much for your helpful comment. We have corrected the errors in the revised manuscript.
Line 192: Why do authors use a colon right after parentheses? I found the same in all figure legends.
Answer: Thank you very much for your helpful comment. We have corrected the errors in the revised manuscript. This is an inappropriate use of punctuation and we have deleted the colons after parentheses in the revised manuscript.
Line 201: Give a citation after the statement.
Answer: Thank you very much for your helpful comment. We have added a citation after the statement in the revised manuscript.
Line 222: In figure 6, please increase the font size of the titles of each figures. It is not legible to read.
Answer: Thank you very much for your helpful comment. We have increased the font size of the titles of each figures in the revised manuscript.
Line 244: In figure 8, adjust the figure position, there are many empty places between figures. Correct the title direction of the figures. Please give in the box of any corner one figure with an enlarged single neuron, so that it would be interesting to see the Ca positive puncta clearly.
Answer: Thank you very much for your helpful comment. We have modified according to the meaningful comments in the revised manuscript.
Line 330: In the last paragraph of the discussion section, please add the limitations of your study and what could be your future plan for translational research, based on your present findings.
Answer: Thank you very much for your helpful comment. We have added the limitations and future plans of the study according to the meaningful comments in the revised manuscript.
Line 388-389: Rewrite the sentence.
Answer: Thank you very much for your helpful comment. We have rewrote the sentence in the revised manuscript.
Line 390-391: Rephrase the sentence.
Answer: Thank you very much for your helpful comment. We have rephrased the sentence in the revised manuscript.
Line 393: Was it a Poly-DL-Lysine coated plate? Mention there.
Answer: Thank you very much for your helpful comment. We have added it in the revised manuscript.
Line 477: In H/R injury procedure, add a reference.
Answer: Thank you very much for your helpful comment. We have added a reference in the revised manuscript.
Line 483: Which cell counting kit did you use? Add details of company information.
Answer: Thank you very much for your helpful comment. We have added the details of company information in the revised manuscript (4.1. Reagents).
Line 488: Which commercial company assay kits did you perform? Again lacking information, please add them.
Answer: Thank you very much for your helpful comment. We have added the details of company information in the revised manuscript (4.1. Reagents).
Line 490: Add the model and company information of the centrifugal machine.
Answer: Thank you very much for your helpful comment. We have added the model and company information of the centrifugal machine in the revised manuscript.
Line 496: Add company information of Fluo-8 AM.
Answer: Thank you very much for your helpful comment. We have added the company information of Fluo-8 AM in the revised manuscript (4.1. Reagents).

Reviewer 2 Report
The manuscript is an interesting mechanistic investigation on the activity of H2S on ROCK2.
The language needs some editing since expressions that are not correct are used through the manuscript. For example bacteria are "transformed" with plasmid, not transferred.
There is no explanation of the term RHNS in the manuscript. I am guessing it means rat hippocampal neurons?
In terms of the results it is not clear to me why the molecular weight of His-tagged ROCK expressed in bacteria is the same as GFP-tagged ROCK in neuronal cells. The GFP tagged should be bigger.
Also why the authors continue with their studies when the write:
After sonication of the lysate, the band was only presented in the precipitation but not supernatant of the transfected E. colis, preliminarily indicating that the His-ROCK2wild was expressed in the transferred E. colis, and the protein may exist in the insoluble form and cannot be further purified. Western-blot results showed that the molecular weight of the ROCK2 expressed in the ROCK2wild-pET-32a(+) transferred E. coli was consistent with that in RHNs. Hence, lysate of the E. colis was used for in vitro phosphorylation assays.
Was the protein found in the soluble fraction?
Or did the author used the protein from the insoluble pellet?
Finally I am not sure how the experiment with the calcium dye worked. The authors are using Fluo-8 dye that has an overlapping spectrum with that of GFP. How can they discriminate between the signal of GFP ROCK from that of the Fluo-8 dye? maybe I missed something here.
Author Response
The manuscript is an interesting mechanistic investigation on the activity of H2S on ROCK2.The language needs some editing since expressions that are not correct are used through the manuscript. For example bacteria are "transformed" with plasmid, not transferred.
Answer: Thank you very much for your helpful comment. We have corrected the errors in the revised manuscript.
There is no explanation of the term RHNS in the manuscript. I am guessing it means rat hippocampal neurons?
Answer: Thank you very much for your helpful comment. RHNs is the abbreviation of rat hippocampal neurons (Line 18 and 81).
In terms of the results it is not clear to me why the molecular weight of His-tagged ROCK expressed in bacteria is the same as GFP-tagged ROCK in neuronal cells. The GFP tagged should be bigger.
Answer: Thank you very much for your helpful comment. In our entire research, we have constructed His, GST, GFP labeled proteins with predicted molecular weight of 170KD, 178KD and 187KD, respectively. In this article, we used His and GFP labeled proteins. We have mislabeled them and have modified them to correct predicted molecular weights in the revised manuscript.
Also why the authors continue with their studies when the write: After sonication of the lysate, the band was only presented in the precipitation but not supernatant of the transfected E. colis, preliminarily indicating that the His-ROCK2wild was expressed in the transferred E. colis, and the protein may exist in the insoluble form and cannot be further purified. Western-blot results showed that the molecular weight of the ROCK2 expressed in the ROCK2wild-pET-32a(+) transferred E. coli was consistent with that in RHNs. Hence, lysate of the E. colis was used for in vitro phosphorylation assays. Was the protein found in the soluble fraction? Or did the author used the protein from the insoluble pellet?
Answer: Thank you very much for your helpful comment. In this study, we found the His-ROCK2wild protein was exist in the insoluble form and cannot be further purified (Revised manuscript Line 90-93). So, lysate precipitation of the E. colis was used for in vitro phosphorylation assays (Revised manuscript Line 95-96).
Finally I am not sure how the experiment with the calcium dye worked. The authors are using Fluo-8 dye that has an overlapping spectrum with that of GFP. How can they discriminate between the signal of GFP ROCK from that of the Fluo-8 dye? maybe I missed something here.
Answer: Thank you very much for your helpful comment. Before calcium imaging, the fluorescence of GFP is taken, and the fluorescence value of GFP will be subtracted when calculating the fluorescence of Fluo-8.

Reviewer 3 Report
In Figure 7 panel A. There is a spare zero.
No major observations
Author Response
In Figure 7 panel A. There is a spare zero.
Answer: Thank you very much for your helpful comment. We have corrected the error in the revised manuscript.

Round 2
Reviewer 1 Report
The authors sufficiently responded to most of the queries. I recommend accepting in the current form.